# Comprehensive Influences of Overexpression of a MYB Transcriptor Regulating Anthocyanin Biosynthesis on Transcriptome and Metabolome of Tobacco Leaves

**DOI:** 10.3390/ijms20205123

**Published:** 2019-10-16

**Authors:** Yuan Zong, Shiming Li, Xinyuan Xi, Dong Cao, Zhong Wang, Ran Wang, Baolong Liu

**Affiliations:** 1Key Laboratory of Adaptation and Evolution of Plateau Biota (AEPB), Northwest Institute of Plateau Biology, The Innovative Academy of Seed Design, Chinese Academy of Sciences, Xining 810008, China; zongyuan@nwipb.cas.cn (Y.Z.); xixy@nwipb.cas.cn (X.X.); caodong@nwipb.cas.cn (D.C.); 2Qinghai Province Key Laboratory of Crop Molecular Breeding, Xining 810008, China; 3University of Chinese Academy of Sciences, Beijing 100049, China; 4National Tobacco Research Center, Zhengzhou Tabacco Research Institute, Henan Zhengzhou 450001, China

**Keywords:** MYB transcriptor, secondary metabolism, anthocyanin, transcriptome, metabolome

## Abstract

Overexpression of R2R3-MYB transcriptor can induce up-expression of anthocyanin biosynthesis structural genes, and improve the anthocyanin content in plant tissues, but it is not clear whether the MYB transcription factor overexpression does effect on other genes transcript and chemical compounds accumulation. In this manuscript, RNA-sequencing and the stepwise multiple ion monitoring-enhanced product ions (stepwise MIM-EPI) strategy were employed to evaluate the comprehensive effect of the MYB transcription factor *LrAN2* in tobacco. Overexpression of *LrAN2* could promote anthocyanin accumulation in a lot of tissues of tobacco cultivar Samsun. Only 185 unigenes express differently in a total of 160,965 unigenes in leaves, and 224 chemical compounds were differently accumulated. Three anthocyanins, apigeninidin chloride, pelargonidin 3-O-beta-D-glucoside and cyanidin 3,5-O-diglucoside, were detected only in transgenic lines, which could explain the phenotype of purple leaves. Except for anthocyanins, the phenylpropanoid, polyphenol (catechin), flavonoid, flavone and flavonol, belong to the same subgroups of flavonoids biosynthesis pathway with anthocyanin and were also up-accumulated. Overexpression of *LrAN2* activated the bHLH (basic helix-loop-helix protein) transcription factor *AN1b*, relative to anthocyanin biosynthesis and the MYB transcription factor *MYB3*, relative to proanthocyanin biosynthesis. Then, the structural genes, relative to the phenylpropanoid pathway, were activated, which led to the up-accumulation of phenylpropanoid, polyphenol (catechin), flavonoid, flavone, flavonol and anthocyanin. The MYB transcription factor *CPC*, negative to anthocyanin biosynthesis, also induced up-expression in transgenic lines, which implied that a negative regulation mechanism existed in the anthocyanin biosynthesis pathway. The relative contents of all 19 differently accumulated amino and derivers were decreased in transgenic lines, which meant the phenylalanine biosynthesis pathway completed the same substrates with other amino acids. Interestingly, the acetylalkylglycerol acetylhydrolase was down-expressed in transgenic lines, which caused 19 lyso-phosphatidylcholine and derivatives of lipids to be up-accumulated, and 8 octodecane and derivatives were down-accumulated. This research will give more information about the function of MYB transcription factors on the anthocyanin biosynthesis and other chemical compounds and be of benefit to obtaining new plant cultivars with high anthocyanin content by biotechnology.

## 1. Introduction

Anthocyanins are glycosylated polyphenolic compounds, and endow flowers, seeds, fruits, and vegetative tissues with a range of colors varying from orange, red, and purple to blue [1]. The intravacuolar environment can influence the color phenotype of anthocyanins because of the characteristics of water-soluble and cell vacuoles location [2]. Over 600 anthocyanins have been identified in nature. The most common anthocyanins are the derivatives of six widespread anthocyanidins, namely, pelargonidin, cyanidin, delphinidin, peonidin, petunidin, and malvidin. Anthocyanins can protect plants against various biotic and abiotic stresses, partially due to their powerful antioxidant properties. In addition, anthocyanin-rich food products have become increasingly popular due to their attractive colors and suggested benefits for human health [3].

Anthocyanins belong to a diverse family of aromatic molecules called flavonoids. Flavonoids contain five major subgroups beside anthocyanins in higher plants: chalcones, flavones, flavonols, flavandiols or proanthocyanidins, and aurones. Most of the structural genes in the anthocyanin biosynthesis pathway have been identified in various plants, such as apple (*Malus domestica*), Arabidopsis (*Arabidopsis thaliana*), grapevine (*Vitis vinifera L.*), and petunia (*Petunia hybrida*) [4,5,6,7]. It is known that the structural genes of anthocyanin biosynthesis contain *phenylalanine ammonia lyase* (*PAL*), *cinnamate 4-hydroxylase* (*C4H*), *4-coumarate-CoA ligase* (*4CL*), *chalcone synthase* (*CHS*), *chalcone isomerase* (*CHI*), *flavanone 3-hydroxylase* (*F3H*), *flavonoid 3-hydroxylase* (*F3′H*) or *flavonoid 3′,5′-hydroxylase* (*F3′5′H*), *dihydroflavonol 4-reductase* (*DFR*), *anthocyanidin synthase* (*ANS*), and *O-methyltransferase* (*OMT*) [8]. The transcription of the structural genes is regulated directly by the MYB-bHLH-WD40 complex (MBW), which is composed of R2R3-MYB, bHLH, and WD40 proteins [9]. the transcriptional activation of MYB transcription factors can usually lead to anthocyanin accumulation in the leaves or fruits in plants [10]. In apples, red fruit color is associated with *MdMYB10* [11], while the purple phenotype of sweet potato is controlled by *IbMYB1* [12]. *LhMYB6* and *LhMYB12* can activate the transcription of the anthocyanin biosynthetic structural genes and cause anthocyanin accumulation in petunia (*Petunia hybrida*) [13].

The overexpression of only MYB transcription factors has been used to enhance anthocyanin content in several plant species. Pattanaik reported that overexpression of the MYB transcription factor *NtAn2* enhanced expression levels of the anthocyanin biosynthetic structural genes and anthocyanin accumulation in tobacco [14], whereas the overexpression of *PtrMYB119* also do the same things in a number of tissues of hybrid poplar [15]. Although it has been proven that the overexpression of MYB transcription factors promotes anthocyanin accumulation in plants, the molecular mechanism has not been clarified in detail. The plant cell is a complex system. The overexpression of a transcription factor will activate structural genes and the expression of structural genes will produce some kind of chemical compound [16]. The plant cell will adapt to changes in the accumulation of chemical compounds within it. Previous researchers have reported on the transcription of a limited number of genes, in response to overexpression of MYB transcription factors, and the accumulation of a limited number of chemical compounds [17,18]. These reports have not clarified the number of genes affected by the overexpression of MYB transcription factors and the turbulence effects of MYB transcription factor overexpression on chemical compounds other than anthocyanin.

Recently, the stepwise multiple ion monitoring-enhanced product ions (stepwise MIM-EPI) strategy was designed to analyze widely targeted metabolomes. The method can quantify hundreds of metabolites simultaneously in rice leaf [19]. Moreover, high-throughput RNA-sequencing (RNA-seq) has emerged as a powerful and cost-efficient tool in providing comprehensive information of the nucleotide sequence and transcript profiling of all genes expressed in the specific tissues of plants [20,21,22]. In this manuscript, these two technologies were employed to uncover the transcript and metabolite differences in the leaves of wild-type (WT) and transgenic lines with high anthocyanin content. It is the first time to reveal the comprehensive function of the MYB transcriptor factor on the transcript and metabolite, except for anthocyanin biosynthesis.

## 2. Results

### 2.1. Phenotype and Anthocyanin Content of Transgenic Line Overexpressing LrAN2

As mentioned previously, *LrAN2* is a MYB transcription factor expressed only in the black fruit of *Lycium ruthenicum Murry* [23]. Overexpression of *LrAN2* causes visible anthocyanin accumulation in various parts of the plant (Figure 1A). The relative anthocyanin contents of roots, stems, leaves, flowers, and seeds were significantly higher in the transgenic lines than the WT (Figure 1B). Although anthocyanin differences are not visible in roots, the contents of this compound were found to be different in the two lines using chemical measurements. Because the anthocyanin content of the leaves was the highest in the three tested transgenic lines, the leaves were chosen for further transcriptome and metabolome analysis.

### 2.2. Chemical Compounds Difference in Transgenic Lines and Wild-Type (WT)

The stepwise MIM-EPI strategy was developed recently to analyze widely targeted metabolomes. The overlapping display analysis of total ion flow charts (TIC) for LC-MS/MS detection and analysis of different quality control samples showed that the repeatability and reliability of the data were good enough to enable further analysis (Appendix A). A total of 639 chemical compounds from 22 classes were detected and 224 compounds from 19 classes exhibited significantly different levels of accumulation in the transgenic lines and WT (Table 1). Transgenic lines displayed higher contents of 109 chemical compounds and the WT showed increased levels of 115 compounds. Although 13 alcohols, two quinones, and five sterides were found, none of these accumulated differently in transgenic lines and the WT (Appendix A).

The average ratio of differently accumulated/total compounds was 35.05%. The ratios of the nine classes were bigger than 35.05% (Table 1). The first seven classes included anthocyanins, polyphenols, flavonols, flavonoids, flavones, flavanones, and phenylpropanoids. All these classes of chemical compounds belong to the flavonoid biosynthesis pathway. The phenylpropanoids, polyphenols, flavones, flavonols, flavonoids, and anthocyanins showed more upregulation in the transgenic lines, except for the compound flavone (Table 1). A total of seven anthocyanins, peonidin O-hexoside, rosinidin O-hexoside, delphinidin, cyanidin 3,5-O-diglucoside (Cyanin), pelargonidin 3-O-beta-D-glucoside (callistephin chloride), peonidin 3-O-glucoside chloride, and apigeninidin chloride, were identified (Appendix A). Seven anthocyanins showed higher levels of accumulation in transgenic lines, while only one accumulated more highly in the WT (Table 1 and Appendix A). The raised accumulation of anthocyanin may be the reason for purple colored leaves in the transgenic lines. Three anthocyanins, including cyanidin 3,5-O-diglucoside, pelargonidin 3-O-beta-D-glucoside, and apigeninidin chloride, were only increased significantly in transgenic lines (Appendix A). The remaining three classes included lipids, vitamins and derivatives, and terpene. The class of compounds containing vitamins and their derivatives showed the same levels of up- and down-regulation (Table 1). Among the lipids, 10 compounds were more highly accumulated in transgenic lines, while 21 were reduced (Appendix A). Interestingly, 19 compounds showing higher levels of production belonged to the lyso-phosphatidylcholines (and derivatives), while 8 downregulated compounds belonged to the octodecanes (and derivatives, Appendix A). Interestingly, all 16 differently accumulated amino acids exhibited lower levels of production in the transgenic lines (Table 1 and Appendix A).

### 2.3. Differently-Expressed Genes in Transgenic Lines and WT

Using transcriptome sequencing, the total number of bases from all samples was over 6 Gb after filtering (Appendix A). The clean reads were further assembled into 160,965 unigenes. The average length of the unigenes was 1233 bp and the length of N50 was 2002 bp (Appendix A). A Blast X search resulted in a total of 135,911 predicted proteins being predicted (Appendix A).

Although a total 160,965 unigenes were found, only 185 were expressed differently in the transgenic lines and WT (Figure 2A, Appendix A). Compared with the WT, 39 unigenes were downregulated and 146 were upregulated (Figure 2A, Appendix A). Fifteen pathways contained only upregulated unigenes in the enriched 30 Kyoto Encyclopedia of Genes and Genomes (KEGG) pathways (Figure 2B). The pathways included amino sugar and nucleotide sugar metabolism, basal transcription factors, cholesterol metabolism, cyanoamino acid metabolism, flavone and flavonol biosynthesis, flavonoid biosynthesis, galactose metabolism, glucosinolate biosynthesis, glutathione metabolism, mitogen-activated protein kinase (MAPK) signaling pathway, nitrogen metabolism, phenylpropanoid biosynthesis, porphyrin and chlorophyll metabolism, starch and sucrose metabolism, and the sulfur relay system. Seven pathways contained only downregulated unigenes. These pathways were lipid metabolism, fatty acid biosynthesis, fatty acid degradation, fatty acid metabolism, monobactam biosynthesis, phagosome assembly, and soluble N -ethylmaleimide-sensitive fusion protein attachment protein receptors (SNARE) interactions in vesicular transport (Figure 2B).

The top thirty upregulated unigenes were homologous to the genes related to anthocyanin biosynthesis (Appendix A), including *ANS, DFR, GST, LrAN2, difF, AN1b, ANP* (anthocyanin permease), *MYB3*, and so on. Apart from being related directly to anthocyanin biosynthesis, *ANP* is a multidrug resistance-associated protein that plays an important role in the transport of anthocyanin pigments into vacuoles [24], while *difF* encodes a cytochrome b5, which is required for full activity of flavonoid 3′,5′-hydroxylase [25]. The log2FoldChange value of the largest upregulated unigene *ANS* reached 9.26 and this was followed by *DFR* (9.00; Appendix A). The total FPKM of the unigenes homologous to *PAL, C4H, 4CL, CHS, CHI, F3H, F3′H, F3′5′H, DFR*, and *ANS* were compared in transgenic lines and the WT. These structural anthocyanin biosynthesis genes were significantly upregulated, except for *4CL*, *F3H*, and *F3′5′H* (Figure 3A, Appendix A). The qPCR experiment confirmed the expression differences between these structural genes (Figure 3B). Undoubtedly, the anthocyanin biosynthesis pathway is activated by the overexpression of MYB transcription factor *LrAN2*. Moreover, 10 upregulated unigenes were homologous to bHLH transcription factor *AN1b*, which has been proven to regulate anthocyanin biosynthesis [26] (Appendix A). Interestingly, ten and one unigenes homologous to transcription factors *MYB3* and *CPC* respectively, showed more expression in transgenic lines (Appendix A).

The anthocyanin biosynthetic pathway and its branches were selected separately, and their structural genes and metabolites were analyzed. Biological reactions with (E)-*p*-coumaric acid as a substrate showed that the expression of these metabolites: Phloretin (Flavanone), Caffeic acid (Phenylpropanoids), Sinapoyl malate (organic acids and derivatives), Myricetin (Flavonol), Cyanidin (Anthocyanins), Pelargonidin (Anthocyanins), in purple tobacco increased at least ten times (measured by log2 value). The expression of structural genes *DFR* and *ANS* in the anthocyanin biosynthesis pathway was the highest among all structural genes, which was consistent with the results of qPCR (Figure 3B).

### 2.4. Conjoint Analysis of Transcriptome and Metabolome

To further evaluate the effects of the transcriptome change on the metabolome, the differently-expressed genes and differently-accumulated compounds were put on the KEGG pathway. Fourteen pathways contained both differently expressed genes and differently accumulated compounds (Figure 4, Appendix A). These were divided further into three classes: the phenylpropanoid secondary metabolism pathway, the amino acid biosynthesis and metabolism pathway, and the ether lipid metabolism pathway. The phenylpropanoid secondary biosynthesis pathway consisted of flavonoid biosynthesis, phenylpropanoid biosynthesis, and flavone and flavonol biosynthesis, while the amino acid biosynthesis and metabolism pathway consisted of monobactam biosynthesis, the sulfur relay system, 2-oxocarboxylic acid metabolism, amino sugar and nucleotide sugar metabolism, glucosinolate biosynthesis, phenylalanine metabolism, cysteine and methionine metabolism, glycine, serine and threonine metabolism, aminoacyl-tRNA biosynthesis, and biosynthesis of amino acids. Only flavonoid biosynthesis could be enriched based on the *p*-value of the transcriptome and metabolome (Figure 4). In the ether lipid metabolism pathway, acetylalkylglycerol acetylhydrolases showed lower expression in the transgenic lines than WT (Appendix A). The acetylalkylglycerol acetylhydrolases may catalyze 1-alkyl-2-acetyl-sn-glycero-3-phosphocholine and H_2_O to produce 1-alkyl-sn-glycero-3-phosphocholine and acetate [27]. 1-alkyl-2-acetyl-sn-glycero-3-phosphocholine is the precursor of lyso-phosphatidylcholine and derivatives, while 1-alkyl-sn-glycero-3-phosphocholine is the precursor of octodecane and its derivatives [27]. The downregulation of the platelet-activating factor acetylhydrolase could explain why 19 lyso-phosphatidylcholines and derivatives were upregulated in transgenic lines and 8 octodecanes and derivatives were produced at lower levels in metabolome analysis.

## 3. Discussion

In this manuscript, we described the effect of *LrAN2* overexpression on the transcriptome and metabolome of tobacco leaves using stepwise MIM-EPI and RNA-seq. These were proven to be effective strategies for analyzing the comprehensive metabolome and transcriptome, respectively. A total of 639 chemical compounds from 22 classes was detected and 160,965 unigenes were assembled during transcriptome analysis.

### 3.1. Transcriptome Turbulence Was Relatively Simple and Metabolome Influence Was Extensive

Interestingly, very few genes showed different levels of expression in transgenic lines and WT using transcriptome analysis. Although many unigenes were assembled, only 185 were shown to be expressed differently, which is a very low number. The top 30 unigenes showing upregulation were related to anthocyanin biosynthesis. These unigenes contained direct structural genes such as *ANS*, *DFR*, and transcription factors related to anthocyanin biosynthesis, but also *ANP* and *difF*, which are accessories that have not been identified in many plants. It could be speculated that overexpression of some regulatory factors with RNA-seq analysis could help in the isolation of structural genes and regulation factors related to specific chemical compound biosynthetic pathways. In this case, a lot of differently expressed unigenes were annotated, particularly upregulated unigenes, but the functions of the remaining unigenes are still unknown. Crispr/Cas9 was used to knock out these differently expressed genes to determine their functions [28].

Of 639 chemical compounds, the contents of 224 were significantly different in the transgenic lines and WT. The ratio of differently accumulated compounds to total identified compounds reached 35.05%. Compared to that of the transcriptome, the influence of the metabolome was extensive.

### 3.2. Overexpression of MYB Transcription Factors Influences Phenylpropanoid Metabolism

In many plants, it has been noted that overexpression of a MYB transcription factor enhances the expression of anthocyanin biosynthetic genes as well as anthocyanin accumulation. This research further confirmed that the chemical classes of phenylpropanoids, polyphenols, flavones, flavonols, flavonoids, and flavanones showed high levels of accumulation (except anthocyanins) in transgenic lines. Moreover, overexpression of *LrAN2* could promote the bHLH transcriptor *AN1b*, which is related to anthocyanin biosynthesis. Transcriptional activation of the MBW complex may regulate the transcription of anthocyanin biosynthetic genes directly. *LrAN2* also promotes the transcription of other transcription factors, homologous to *MYB3* and *CPC.* The transcription factor *MYB3* is believed to regulate proanthocyanin (epiafzelechin and derivatives) biosynthesis [29], while *CPC* negatively regulates anthocyanin biosynthesis in Arabidopsis as it competes with the R2R3-MYB transcription factor *PAP1/2* [30]. The accumulation of epiafzelechin and derivatives in the transgenic lines could be explained by the up-expression of *MYB3*. The upregulation of *CPC* due to anthocyanin accumulation is a mechanism for reducing anthocyanin biosynthesis.

### 3.3. Amino Acid Biosynthesis is Influenced by the Overexpression of MYB Transcription Factor

Phenylalanine is the substrate for the phenylpropanoid biosynthesis pathway, and it is an amino acid. Undoubtedly, phenylalanine competes for the same substrates as other amino acids. The relative contents of all 16 differently accumulated amino acids and derivatives were decreased in the transgenic lines. In conjoint analysis of the transcriptome and metabolome, apart from the phenylpropanoid secondary biosynthesis pathway and fatty acid degradation pathway, the enriched KEGG pathways exhibiting different gene expression and chemical compound levels were related to amino acid biosynthesis and metabolism. The cyanoamino acid metabolism pathway derived from phenylalanine was active in the transcriptome. Although a chemical difference has not been found, this metabolic pathway is connected directly to those of glycine, serine and threonine, cystine and methionine, glutathione, and other amino acids.

This research describes a relatively comprehensive change in the metabolome and transcriptome of tobacco leaves following overexpression of a MYB transcription factor. Six transgenic tobacco strains overexpressing *LrAN2* were obtained. This study provides more information about the function of MYB transcription factors in anthocyanin biosynthesis and that of other chemical compounds. These results will help researchers to establish new plant cultivars containing high levels of anthocyanin through transgenic technology.

## 4. Materials and Methods

### 4.1. Materials

The *Nicotiana tabacum L.* cultivar Samsun was chosen as a transformation plant. All plant materials were stored in the Northwest Plateau Institute of Biology, Chinese Academy of Sciences. Samsun served as the control group and transgenic tobacco served as the experimental group. Each group was repeated three times. Transcriptome sequencing and metabolome sequencing used the same tobacco leaf. Chemical and molecular regents were purchased from Gansu Pengcheng Biological Technology Development Company.

### 4.2. Obtaining Transgenic Lines with Overexpression of MYB Transcription Factor

The MYB transcription factor *LrAN2* was first reported by our laboratory. It expressed only in the black fruit of *Lycium ruthenicum Murry* and the allele variation of *LrAN2* is strictly related to the black fruit trait [23]. In this study, *LrAN2* was transformed into tobacco Samsun to obtain transgenic lines with high anthocyanin content. The vector PJAM1502 with a double 35 s promoter was used for the transformation. The construct PJAM1502:*LrAN2* was established using a Gateway Cloning Kit (Invitrogen, Carlsbad, CA, USA). The freeze-melt method was employed to transform the binary vectors into *Agrobacterium tumefaciens* strain GV3101. The leaf disc transformation method was used for the tobacco transformation [31]. Regeneration tissues were grown on selective media (0.7% (w/v) agar, 3% (w/v) sucrose, 1.0 mg/L 1-naphthaleneacetic acid (NAA), 1.0 mg/L 6-benzylaminopurine (6-BA), and 150 mg/L kanamycin. The positive shoots were planted in the greenhouse under long-day lighting (16 h light/8 h dark). Though 23 positive lines were obtained, only three T3 family lines carrying the objective gene without segregation of character were selected for further experiments.

### 4.3. Anthocyanin Content Measurement

The “Total Monomeric Anthocyanin Pigment Content of Fruit Juices, Beverages, Natural Colorants, and Wines” (AOAC (Association of Official Analytical Chemists) Official Method 2005.02) method was used to measure relative anthocyanin content. HCl (1% v/v) was added to 100 mg of comminuted plant tissues (root, steam, leaf, seed, and flower), and the mixtures were incubated at 4 °C overnight in the dark to extract anthocyanin, with three repetitions in each plant.

### 4.4. Analysis of Chemical Contents

Transgenic lines and WT were cultured in a plant incubator for 1 month. The leaves were stripped and stored in a refrigerator at −80 °C before chemical and transcriptome analyses. Freeze-dried leaves were crushed with a mixer mill (MM 400, Retsch) at 30 Hz with a zirconia bead for 1.5 min. 100 mg powder was weighed and extracted overnight with 1.0 mL 70% aqueous methanol at 4 °C. After centrifugation at 10,000× *g* for 10 min, the extracts were filtered by the 0.22 μm filter membrane (ANPEL, Shanghai, China, http://www.anpel.com.cn/).

An LC-electrospray ionization-tandem MS (ESI MS/MS) system (high-performance LC (HPLC), Shim-pack UFLC SHIMADZU CBM30A system, www.shimadzu.com.cn/; MS, Applied Biosystems 6500 Q TRAP, www.appliedbiosystems.com.cn/) was used to analyze the sample extracts (HPLC: column, Waters ACQUITY UPLC HSS T3 C18 (1.8 μm, 2.1 mm*100 mm); solvent system: water (0.04% acetic acid); acetonitrile (0.04% acetic acid); gradient program, 95:5 v/v at 0 min, 5:95 v/v at 11.0 min, 5:95 v/v at 12.0 min, 95:5 v/v at 12.1 min, 95:5 v/v at 15.0 min; temperature, 40 °C; injection volume: 2 μL; flow rate, 0.40 mL/min). Then, the effluent was alternatively connected to an ESI-triple quadrupole-linear ion trap (Q TRAP)-MS.

A triple quadrupole-linear ion trap mass spectrometer (Q TRAP, API 6500 Q TRAP LC/MS/MS system) was used to acquire Linear ion trap (LIT) and triple quadrupole (QQQ) scans. The triple quadrupole-linear ion trap mass spectrometer was equipped with an ESI turbo ion-spray interface, operated in a positive ion mode and controlled by Analyst 1.6.3 software (AB Sciex, Boston, USA). The ESI source operation parameters were as follows: ion source, turbo spray, ion source gas I (GSI), gas II (GSII), and curtain gas (CUR) were set to 55, 60, and 25.0 psi, respectively. The collision gas (CAD) was high, ion spray voltage (IS) 5500 V, source temperature 500 °C. In QQQ and LIT modes, 10 and 100 μmol/L polypropylene glycol solutions were used in Instrument tuning and mass calibration, respectively. QQQ scans were acquired as multiple reaction monitoring (MRM) experiments (the collision gas (nitrogen): 5 psi). Declustering potential (DP) and collision energy (CE) for individual MRM transitions was done with further DP and CE optimization. A specific set of MRM transitions were monitored for each period according to the metabolites eluted within this period.

### 4.5. Illumina Sequencing and Data Analysis

Total RNA was extracted with an RNAprep Pure Plant Kit (Tiangen Company, Beijing, China). The quality of the total RNA was checked by electrophoresis and the concentration was determined by NanoDrop (Thermo Scientific, Wilmington, DE, USA). The cDNA libraries were created according to the standard preparation method for mRNA-seq samples (Illumina Inc, San Diego, CA, USA).

Using an Illumina HiSeq 2000 instrument (Illumina Inc., San Diego, CA, USA), the DNA library was sequenced by Novogene with three repeats. The original reading was filtered to remove ambiguous, joint and low-quality sequences before sequence assembly. After purity filtration, de novo assembly of the transcriptome into unigenes was carried out using Trinity, a short-read assembly program [32]. The unigene sequences were compared to protein databases (Nr, Swiss-Prot, KEGG, and Clusters of Orthologous Groups of proteins (COG)) with software Blast X (v2.2.25, National Center for Biotechnology Information, Bethesda, Maryland, USA) (e-value < 0.00001). Based on the blast results, the CDS from unigene sequences were extracted and translated into peptide sequences. The blast results were also used in training ESTScan [33]. The expression levels of the unigenes were calculated based on the FPKM (fragments per kb per million reads) value. The difference in unigenes between purple and green leaf transcripts was analyzed by IDEG6 software (1.0, BGI, ShenZhen, GuangDong, China) [34]. The false discovery rate (FDR) method was introduced to determine the threshold *p*-value at FDR ≤ 0.001, | log2 ratio > 1 was determined by the significance of the differential expression of unigenes. All differentially abundant unigenes were mapped to the Gene ontology (GO) and KEGG databases using Blast X (e-value < 0.00001), and the respective numbers of unigenes for every GO and KEGG orthology (KO) term were calculated. Based on the hypergeometric test, significantly enriched GO and KO terms were identified to compare these unigenes with the whole transcriptome background [35].

### 4.6. Quantitative Reverse-Transcription-PCR

The cDNA was synthesized from the same total RNA for Illumina sequencing using a reverse transcription kit (Thermo Fisher First Strand cDNA Synthesis Kit, Beijing, China). The β-ACTIN gene was selected for cDNA normalization and several differently expressed genes were selected to confirm the results of RNA-seq. The primers for the selected genes were designed by Primer 5.0 (Appendix A). The qPCR was conducted with the Premix Ex Taq™ probe (TaKaRa, China) in Applied Biosystems QuantStudio3 (Thermo Fisher Company, Beijing, China). The thermal cycle for qPCR was 95 °C for 30 s, followed by 40 cycles of 95 °C for 5 s, and 60 °C for 34 s. The last stage was 95 °C for 15 s, 60 °C for 1 min, and 95 °C for 15 s. Each plate was repeated three times. The 2-ΔΔCT method was used to analyze the expression levels of the different genes [36].

### 4.7. Statistical Analysis

All statistical analyses were carried out using the software package SPSS (22.0, IBM SPSS Statistics, Chicago, Illinois, USA) for Windows 11.5 [37,38]. The method was UNIANOVA, and the POSTHOC was DUNCAN ALPHA (0.05).

## 5. Conclusions

The MYB transcription factor *LrAN2* was first reported by our laboratory. It expressed only in the black fruit of *Lycium ruthenicum Murry* and the allele variation of *LrAN2* is strictly related to the black fruit trait [23]. Overexpression of *LrAN2* activates the pathway of anthocyanin synthesis and metabolism in tobacco. Only 185 unigenes were expressed differently in leaves and 224 chemical compounds exhibited differences in accumulation. Three anthocyanins lead to the purple leaf phenotype. The main pathways of flavonoid biosynthesis were upregulated. This research provides more information about the function of MYB transcription factors in anthocyanin biosynthesis and the production of other chemical compounds. This work will help breeders to obtain new plant cultivars with high anthocyanin contents using biotechnology.

## Figures and Tables

**Figure 1 ijms-20-05123-f001:**
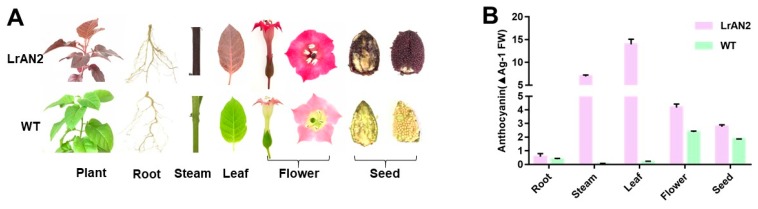
Phenotype and anthocyanin content in transgenic lines and wild-type (WT). (**A**) The phenotype of roots, stems, leaves, seeds, and flowers of transgenic lines and WT. (**B**) The anthocyanin content in different tissues of transgenic lines and WT.

**Figure 2 ijms-20-05123-f002:**
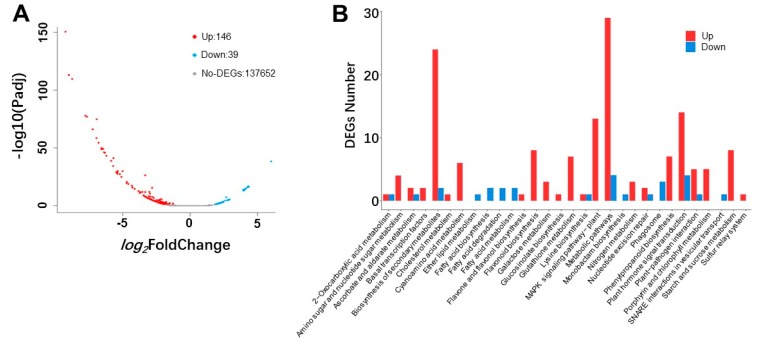
Differentially expressed unigenes (DEGs) in transgenic lines and WT. (**A**) The volcano distribution map of all DEGs. The X-axis represents the FoldChange in expression after conversion of the log2 value. The Y-axis represents the value of -log10 (Padj). The blue point represents the upregulated genes (log2FoldChange ≥ 1, Padj ≤ 0.05), 146 unigenes expressed in transgenic lines were higher than the control. The red point represents the downregulated genes (log2FoldChange ≤ −1, Padj ≤ 0.05), 39 unigenes expressed in transgenic lines were lower than the control. The gray point represents no DEGs. 137,652 unigenes did not express differently in the transgenic lines and WT. (**B**) Enrichment Pathway Map of DEGs.

**Figure 3 ijms-20-05123-f003:**
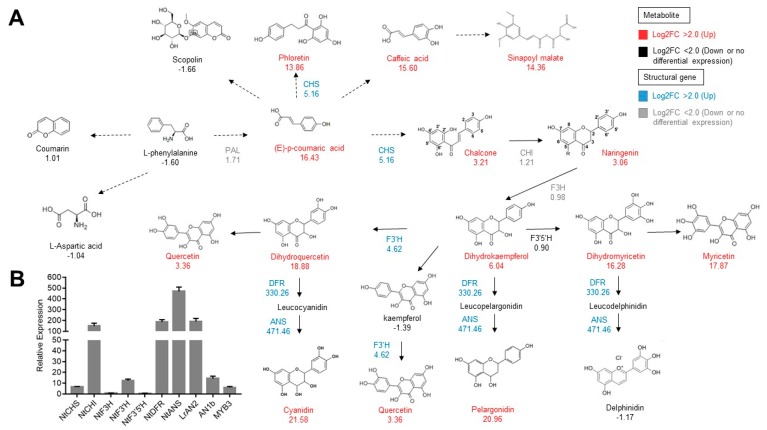
The relative expression levels of the unigenes related to anthocyanin biosynthesis. (**A**) The biosynthesis pathway. Arrows show the metabolic stream, left or upward arrows represent the genes catalyzing the progress, the blue abbreviations express the genes found in the assembly unigenes, and the number under the genes represents the relative expression level in transgenic lines versus the WT. The numbers under the chemical compounds represent the relative contents of the compounds accumulated in the transgenic lines versus the WT. These structural anthocyanin biosynthesis genes were more significantly upregulated in transgenic lines than WT. (**B**) The relative transcript level of the genes relative to anthocyanin biosynthesis based on qPCR experiments.

**Figure 4 ijms-20-05123-f004:**
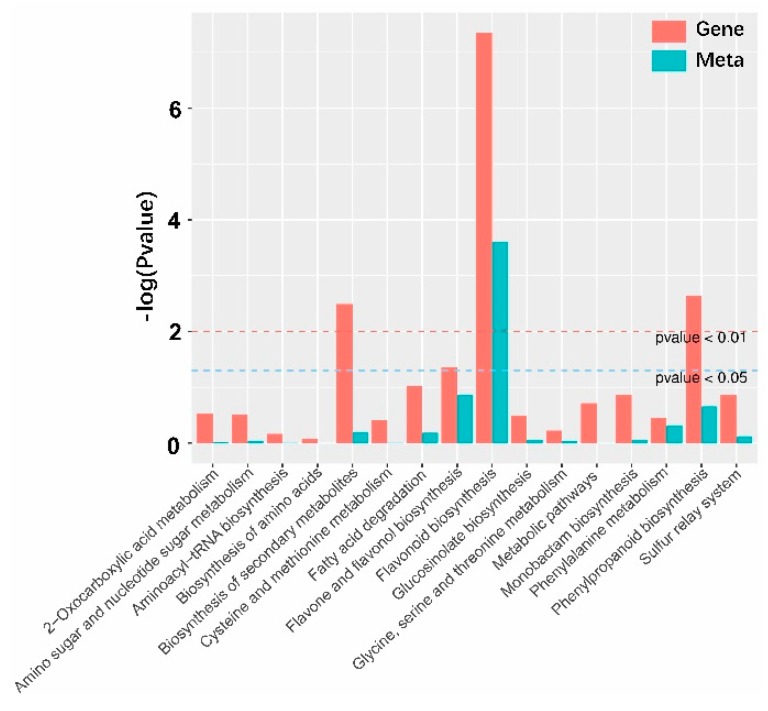
The Kyoto Encyclopedia of Genes and Genomes (KEGG) enrichment histogram of conjoint analysis of differently expressed genes and accumulated metabolites. The X-axis represents the metabolic pathways, the Y-axis represents the expression as -log (*p*-value). The red columns represent the enrichment p-values of differentially expressed genes, and the green columns represent the enrichment p-values of different metabolites.

**Table 1 ijms-20-05123-t001:** Summary of differently accumulated chemical compounds in the transgenic lines and WT.

Category	Total Compounds	Differently Accumulated Compounds	Percent (%)	Up-Accumulation Compounds	Down-Accumulation Compounds
Anthocyanins	9	7	77.78	6	1
Polyphenol	12	8	66.67	7	1
Flavonol	29	17	58.62	12	5
Flavonoid	18	10	55.56	8	2
Flavone	49	27	55.10	11	16
Flavanone	19	9	47.37	7	2
Phenylpropanoids	53	25	47.17	18	7
Lipids	68	31	45.59	10	21
Vitamins and derivatives	15	6	40.00	3	3
Terpene	17	6	35.29	2	4
Organic acids and derivatives	97	34	35.05	17	17
Phenolamides	16	5	31.25	1	4
Others	23	5	21.74	1	4
Amino acid and derivatives	82	16	19.51	0	16
Nucleotide and derivates	47	8	17.02	4	4
Isoflavone	6	1	16.67	0	1
Carbohydrates	19	3	15.79	1	2
Alkaloids	33	5	15.15	1	4
Indole derivatives	7	1	14.29	0	1
Alcohols	13	0	0.00	0	0
Quinones	2	0	0.00	0	0
Sterides	5	0	0.00	0	0
Total	639	224	35.05	109	115

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
