# Peer review of "Comprehensive Influences of Overexpression of a MYB Transcriptor Regulating Anthocyanin Biosynthesis on Transcriptome and Metabolome of Tobacco Leaves"

_ijms, 2019, doi:10.3390/ijms20205123_

Round 1
Reviewer 1 Report
The manuscript “Comprehensive influences of overexpression of a MYB transcriptor regulating anthocyanin biosynthesis on transcriptome and metabolome of tobacco leaves” by Yuan Zong and colleagues studied provide data about the influence of overexpression of transcription factors on the activation of structural genes for anthocyanin biosynthesis and accumulation in tissues of tobacco leaves. The topic of the manuscript its interesting and give us information about the individual anthocyanins biosynthesis and the factors that could determine their accumulation in tobacco leaves. The manuscript its clear, well written, the methodologies are appropriate, and the references used are in general adequate. Thus, in my opinion, there are only a few minor points that should be revise before publication.
Comments:
1 - I suggest to rewrite the abstract. Include only the major results and the main topics. Some sentences are not relevant.
2 - Introduction, line 8-9: There are contradictory results about the antioxidants capacity of anthocyanins. Introduce references about this topic.
3 - Introduction, line 17: Grapes not only from Vitis vinifera. Also from no Vitis vinifera species. Revise it.
4 - Introduction, lines1-2, last sentences: This is results. I suggest to delete and make it clear how relevant and innovative the work is.
5 - Results, line 36: Authors didn’t detected and quantified malvidin ? In some plant issues, it’s the major individual anthocyanin detected.
6 - Results, lines 4; 33, page 6 (and others). I suggest to write “p” in italic form.
7 - Material and methods, lines 20-21, item 4.3: Introduce an independent topic about the statistical analysis. Introduce also the version of the SPSS program used and the Statistical test used.
Author Response
Response to Reviewer 1 Comments
Comments:
1 - I suggest to rewrite the abstract. Include only the major results and the main topics. Some sentences are not relevant.
Response: The abstract had been rewritten.
2 - Introduction, line 8-9: There are contradictory results about the antioxidants capacity of anthocyanins. Introduce references about this topic.
Response: The references had been added.
3 - Introduction, line 17: Grapes not only from Vitis vinifera. Also from no Vitis vinifera species. Revise it.
Response: The “Grapes” had been changed as “Grapevine”
4 - Introduction, lines1-2, last sentences: This is results. I suggest to delete and make it clear how relevant and innovative the work is.
Response: The results had been deleted, and the innovative points had been emphasized.
5 - Results, line 36: Authors didn’t detected and quantified malvidin? In some plant issues, it’s the major individual anthocyanin detected.
Response: The malvidin didn’t been quantified.
6 - Results, lines 4; 33, page 6 (and others). I suggest to write “p” in italic form.
Response: The “p” had been rewritten in italic form.
7 - Material and methods, lines 20-21, item 4.3: Introduce an independent topic about the statistical analysis. Introduce also the version of the SPSS program used and the Statistical test used.
Response: The introduction about SPSS had been added.
Reviewer 2 Report
The manuscript presents some shortcomings and inaccuracies, in particular regarding Materials and Methods, and for this reason it requires a major revision.
In fact:
2 line 36: please add “in detail” after “the molecular mechanism has not been clarified”; 2 lines 39-44: please add references; 3 line 8: Please write the full Latin name of L. ruthenicum in italics “Lycium ruthenicum” adding also “Murray” or “Murr.” because abbreviation is not opportune having the MS a different plant subject; Figure 1: please enlarge placing A and B in a column; Figure 2 and 3: as for Figure 1; Figure 3: please add comments concerning the trend of gene expression and of products; 8 lines 1-3: references are necessary; 8 lines 35-40: please be more specific concerning how obtain new cultivars, do you mean OGM? 8 line 43: please correct in “Nicotiana tabacum L.”; 9 line 8: please use Italic for Agrobacterium tumefaciens; Main points. Along all MS Authors have compared transgenic lines (in a not specified number, but more than one) with (one) WT: that in principle is not correct: Therefore, Authors must explain this point, indicate clearly how many transgenic line were employed, and describe the transgenic line (or lines) employed. Moreover, Authors must explain the meaning of the phrase at lines 17-18 (“method was used to measure relative anthocyanin content in three independent experiments”) and how many plants / type they used for chemical analysis and RNA extraction; Conclusion line 40 of pag. 10: please clearly indicate that LrAN2 derives from Lycium ruthenicumAuthor Response
Response to Reviewer 2 Comments
2 line 36: please add “in detail” after “the molecular mechanism has not been clarified”;
Response: “in detail” had been added.
2 lines 39-44: please add references;
Response: The references had been added.
3 line 8: Please write the full Latin name of L. ruthenicum in italics “Lycium ruthenicum” adding also “Murray” or “Murr.” because abbreviation is not opportune having the MS a different plant subject;
Response: The Latin name had been revised.
Figure 1: please enlarge placing A and B in a column; Figure 2 and 3: as for Figure 1;
Response: A and B had been enlarged in these figures.
Figure 3: please add comments concerning the trend of gene expression and of products;
Response: The comments had been added.
8 lines 1-3: references are necessary;
Response: The references had been added.
8 lines 35-40: please be more specific concerning how obtain new cultivars, do you mean OGM?
Response: New cultivars could be obtained through OGM. We had added the specific description in the segment.
8 line 43: please correct in “Nicotiana tabacum L.”;
Response: It had been corrected.
9 line 8: please use Italic for Agrobacterium tumefaciens;
Response: It had been corrected.
Main points. Along all MS Authors have compared transgenic lines (in a not specified number, but more than one) with (one) WT: that in principle is not correct: Therefore, Authors must explain this point, indicate clearly how many transgenic line were employed, and describe the transgenic line (or lines) employed. Moreover, Authors must explain the meaning of the phrase at lines 17-18 (“method was used to measure relative anthocyanin content in three independent experiments”) and how many plants / type they used for chemical analysis and RNA extraction;
Response: Thanks for this suggestion. Three transgenic line were employed for anthocyanin content measurement, chemical contents analysis, and transcriptome analysis, which had been descripted in the manuscript.
Conclusion line 40 of pag. 10: please clearly indicate that LrAN2 derives from Lycium ruthenicum
Response 11: The relative information had been added.
Round 2
Reviewer 2 Report
Pag. 3, line 13: Authors must write “.. higher in the three tested transgenic lines ..” to indicate immediately to the readers that three lines were analised;
Pag. 9 line 19-20: “Through 23 positive lines were obtained, only three T3 family lines carrying the objective gene without separation were selected for further experiments”, Authors must specify the meaning of “without separation”
Pag. 9 line 24: the phrase “method was used to measure relative anthocyanin content in three independent experiments” was not corrected, so Authors must explain how many different plants they used for chemical analysis and RNA extraction, or modify the phrase.
Author Response
Response to Reviewer 2.2 Comments
Pag. 3, line 13: Authors must write “.. higher in the three tested transgenic lines ..” to indicate immediately to the readers that three lines were analised;
Response: Thanks for this suggestion. This statement had been added.
Pag. 9 line 19-20: “Through 23 positive lines were obtained, only three T3 family lines carrying the objective gene without separation were selected for further experiments”, Authors must specify the meaning of “without separation”
Response: Thanks for this suggestion. It had been revised, without separation means without segregation of character
Pag. 9 line 24: the phrase “method was used to measure relative anthocyanin content in three independent experiments” was not corrected, so Authors must explain how many different plants they used for chemical analysis and RNA extraction, or modify the phrase.
Response: Thanks for this suggestion. We had revised the phrase and added three repetitions in each plant.